# DOMAIN-GROUNDING OF NEURAL NETWORKS FOR SPATIOTEMPORAL REASONING

## ABSTRACT

Neural Networks are powerful approximators for learning to reason from raw data (e.g., pixels, text) in spatio-temporal domains (e.g., traffic-scene understanding). However, several recent studies have shown that neural networks are prone to erroneous or sometimes absurd *reasoning* that lacks *domain-grounding* (e.g., adhering to intuitive physics and causality). Incorporating comprehensive symbolic representation for domain understanding as part of a consolidated architecture offers a promising solution. In this paper, we take a dynamical systems perspective of a neural network and its training process, and formulate domain knowledge-dependent constraints over its internal structures (parameters and inductive biases) during training. This is inspired by *control barrier function*, a constraint specification method from control theory. In particular, we specify the domain knowledge using Knowledge Graphs in our approach. To demonstrate the effectiveness of our approach, we apply it to two benchmark datasets focused on spatiotemporal reasoning: CLEVRER and CLEVRER-Humans, both centered around the task of question answering. Furthermore, we propose novel ways to evaluate if *domain-grounding* is achieved using our method. Our results show that the proposed methodology improves domain-grounding and question-answering accuracy while endowing the model with enhanced interpretability - an interpretability score that specifies to which extent the domain constraints are followed or violated.

## 1 INTRODUCTION

Neural networks (NNs) have transformed how researchers, businesses, and consumers integrate Artificial Intelligence (AI) into their daily operations. Despite their impressive capabilities, contemporary NNs (e.g., Large Language Models) have consistently exhibited significant concerns, including failures in elementary physics or mathematics tasks and common-sense reasoning (Frieder et al., 2023). Additionally, NNs tend to exhibit hallucinations, confidently producing information that appears fluent and eloquent but lacks grounding in reality or truth (Shen et al., 2023). We will refer to this as a lack of *domain-grounding*. Consequently, many researchers argue that NNs excel in tasks like data modality compression and pattern recognition (e.g., language and vision) but fall short in terms of *actual reasoning*, a concept that remains vaguely defined with anecdotal examples (Davis, 2023). In this work, we investigate the capabilities of NNs in spatiotemporal question answering, which requires a domain-grounded understanding of spatiotemporal concepts and the ability to draw conclusions based on this understanding - which we refer to as "reasoning".

To this end, we begin by explaining how NN learning can be framed as a problem of controlling a dynamical system. We then draw inspiration from a concept in control theory called *control barrier functions* (CBFs) to formulate domain-grounding constraints in the context of NN learning (Ames et al., 2019). While traditional control theory places constraints on physical parameters like position and velocity, the vast number (billions) of parameters in NN learning makes defining constraints impractical, often leading to an intractable set of constraints. Therefore, we use symbolic structures like Knowledge Graphs (KGs) to define a domain model and constraints in a compact form. Specifically, we derive graph structured abstractions from the NN's parameters and "control" the NN's learning by minimizing the error between these derived graph structured abstractions and the KG representing the domain model constraints. Figure 1 shows a detailed depiction of this concept.

***Why KGs?*** KGs are a good choice of knowledge representation for many reasons. They are proven for constructing *scalable*, *robust*, and *sufficiently expressive* domain models that transcend specific datasets and find relevance in diverse domains (Jaimini et al., 2022). In addition to their adaptability across domains, KGs enable storing high-level and granular (low-level) information essential for learning pipelines, particularly useful in real-world problem-solving. This enables quantitative and qualitative reasoning, making them an attractive choice for "machine-comprehension" of intricate phenomena. KGs also allow for relatively easy extensibility when adding new domain knowledge, making them scalable and adaptive. Finally, there is a rich and highly-developed infrastructure for operating with KGs in both academia and industry, making them a pragmatic choice for placing constraints on NN learning pipelines (Yahya et al., 2021; Hogan et al., 2021).

**Related Work:** Past work has attempted to achieve domain-grounding in NNs by leveraging the *in-context learning* abilities of modern NNs (Brown et al., 2020). In-context learning refers to the ability of NNs to adapt by using user-specified input context and instructions, an ability achieved using instruction tuning and alignment (Gupta et al., 2022). These works provide exemplars of positive and negative domain-grounding through few-shot prompting, a framework now recognized as the *pre-train, prompt, and predict* paradigm. Moreover, researchers have introduced improvements to prompting methods, including chain-of-thought (CoT), tree-of-thought, and graph-of-thought promptings (Wen et al., 2023). These enhancements aim to elucidate the NN's "thought process" when arriving at an outcome, for instance, by appending *think step-by-step* to the prompt. This helps in evaluating the NN's efficacy in achieving domain-grounding. Although prompting emulates the "thought processes," perhaps similar to how humans reason, there are two primary issues: (1) Producing a series of intermediate computational traces that leads the model to a specific outcome, does not mean that the NN is actually "reasoning." (2) Even if the NN is actually "reasoning," there is no guarantee of adherence to a domain model grounded in reality. For example, the NN could come up with what "it thinks" is a well-reasoned physical process for placing a solid object on top of a soap bubble. The occurrence of the second issue is a direct consequence of the first, underscoring the need for methods with domain-grounding guarantees during NN training. For example, Wei et al. (2022) leveraged CoT prompting with PALM 540B for task decomposition. Despite their reported success, using prompting to generate fine-grained reasoning steps did not always yield consistent results using different versions of GPT-4 (Chowdhery et al., 2022; Chen et al., 2023; OpenAI, 2023). The same work also indicates that, even when reasoning steps are correctly reproduced, they do not always match the model selecting the correct solution/answer to a problem/question.

**Main Contribution:** The main contributions of our work are (1) We provide a control-theoretic perspective for NN learning with KGs for representing domain models and constraints; (2) We present a systematic approach for achieving domain-grounded NN learning; and (3) We provide theoretical analysis and empirical evidence using the benchmark datasets CLEVRER and CLEVRER-Humans, centered around spatiotemporal question answering, that demonstrates the effectiveness of the proposed approach.

## 2 BACKGROUND

### 2.1 MOTIVATING EXAMPLE - CONSTRAINED CONTROL OF DYNAMICAL SYSTEM

Consider a model of a drone, represented as a dynamical system governed by

$$z(t+1) = F(z(t), d(t), \omega(t)), \quad d(t+1) = f(z(t), d(t), \omega(t)), \quad d(0), f(\cdot) \in D, \quad (1)$$

where $t = 0, 1, \ldots, z(t)$ represents the latent state of the drone (e.g., pitch, yaw, roll angles, coordinate positions, and angular rates), $F$ denotes the dynamics of the state, $\omega(t)$ models some exogenous inputs influencing the states, $d(t)$ denotes the position of the drone at time $t$, and $f$ is an observable function. The symbol $D$ denotes a set of feasible (next) positions of the drone, typically constraining the state of the drone to be within a safe region of operation. For example, in a very simplistic case, we may define the position of the drone along the horizontal axis with a linear rate of change, i.e., $d(t+1) = ad(t) + \omega(t)$ for some $\omega(t), a \geq 0$. This recursive relation, along with the constraint in equation 1 demands that the drone is only allowed to move "forward" ($a, \omega(t) \geq 0$) and that its next position can never be outside the set $D$, i.e., $d(t+1) \in D$. Now consider that the drone is only allowed to be in positions $D = [0, d_{max}]$, where $d_{max} > 0$. The value $d_{max}$, for example, could denote the position of an obstacle that the drone cannot cross even though physical positions

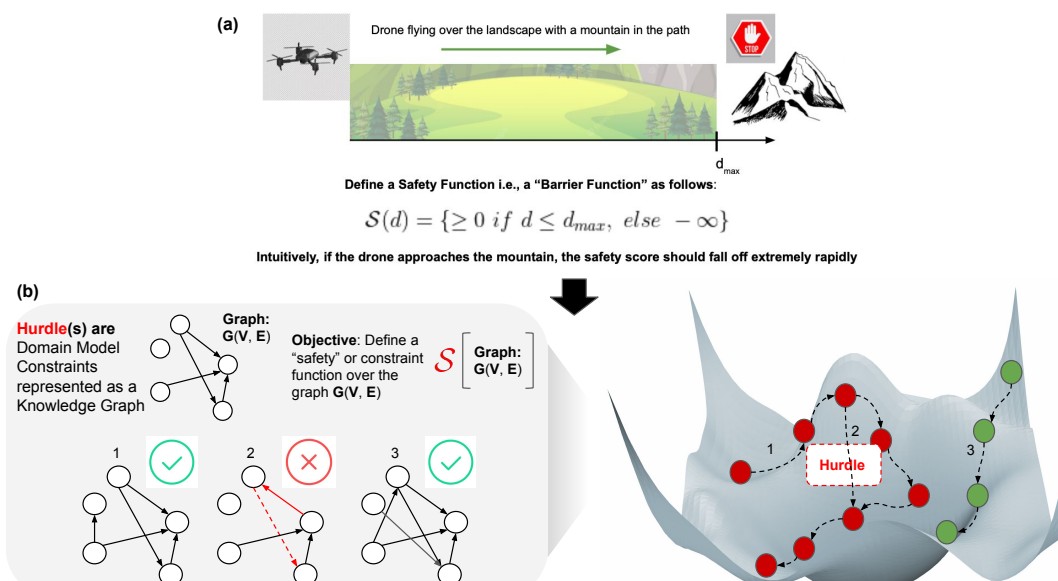

Figure 1: **Conceptual Flow in the Paper** - First, we describe a basic dynamical system, a drone motion system, and then show how concepts in drone control can be extended to constrain NNs where the constraints are specified as KGs **(a)** Illustrates how drone motion safety (avoiding the mountain in its path) is formulated in control theory as a function of its physical parameters (its position $d$ in the figure). This is known as a *Control Barrier Function* **(b)** (right) Illustrates how a neural network's parameters moving on its loss landscape during training can be likened to drone control and consequently be directed to avoid "hurdles" (bad paths on the surface, e.g., the path labeled "2" in the figure) analogous to a mountain **(b)** (left) Illustrates how we propose to formulate safety as a function of graph structured abstractions derived from the NNk's parameters instead of on the parameters themselves. This is due to the difficulty in specifying domain constraints at the level of parameters, which is more natural to specify as a KG. We can train the NN to respect the domain constraints by minimizing the error between the graph abstractions derived from its parameters and the KG in an end-to-end optimization framework.

beyond that hurdle exist, i.e., the positions $d \geq d_{max}$ are *reachable* positions for the drone. These constraints typically do not form an intrinsic component of the drone's dynamic model; instead, they arise from environmental or domain-specific information. Ensuring the drone's safe control thus involves devising an external control signal that guides the drone effectively while adhering to the constraints imposed on its observable parameters.

## 2.2 LEARNING TO CONTROL WITH CONSTRAINTS USING CBFS

If we do not have access to the dynamics of the drone position or it's rate of change along the horizontal axis with time, we may use the observed data ($d(t)$) at different points in time to learn an approximation of the dynamics given as $x(t+1) = \alpha \cdot x(t) + \beta \omega(t)$ such that $||d(t+1) - x(t+1)||$. Then the the drone can be steered by learning $\omega(t)$ as a function of $x(t)$. Additionally, we must ensure that the learned $x(t)$ and $d(t)$ are steered using $\omega(x(t))$ in a desired manner while constrained to take values such that $x(t), d(t) \in [0, d_{max}]$. We can enforce this constraint while learning a control signal $\omega(x(t))$ by formulating this as a penalty $\mathcal{S}(d(t)) = \{0, \text{ if } d(t) \leq d_{max}, \text{ else } \infty\}$. Thus, the learning objective now requires learning the $\omega(t)$ that leads to the minimum value of $||d(t) - d_{target}|| + \mathcal{S}(d(t))$. The function $S(d(t))$ is referred to as the CBF. The quantity $||d(t) - d_{target}|| + \mathcal{S}(d(t))$ is not differentiable by definition of $\mathcal{S}(d(t))$. Hence, a continuous reformulation of the penalty function $\mathcal{S}(d)$ is often used, which is characterized by $\frac{\partial \mathcal{S}(d(t))}{\partial t} \geq -\gamma \mathcal{S}(d(t)), \gamma \geq 0$ for all $d(t) \in D$ Ames et al. (2019).

Now, we can define a constrained minimization objective as:

$$\min_{\omega} ||d(t+1) - d_{target}|| + \mathcal{S}(d(t)), \quad d(t+1) = f(z(t), d(t), \omega(t)), \quad d(0) \in D,$$
$$\text{such that } \frac{\partial \mathcal{S}(d(t))}{\partial t} \geq -\gamma \mathcal{S}(d(t)), \; \gamma \geq 0. \tag{2}$$

This objective has a solution in CBF theory and has the following salient features:

**Forward Invariance** If $d(t)$ is initialized such that $d(t) \in D$, i.e., the constraint is satisfied, then the solution guarantees *forward invariance*, i.e., all $d(t+1)$ also satisfies the constraint defined by $\mathcal{S}(d(t))$.

**Stable Solution for General Classes of Dynamical Systems** In general, stable solutions exist for the optimization of certain classes of systems subject to barrier function constraints, e.g., for non-linear control affine dynamical systems described as follows:

$$\min_{\omega} ||(\omega(t) \cdot g(d(t)) + f(d(t)) - d_{target}|| + \mathcal{S}(d(t)),$$
$$\text{such that } \frac{\partial \mathcal{S}(d(t))}{\partial t} \geq -\gamma \mathcal{S}(d(t)), \; \gamma \geq 0. \tag{3}$$

Thus, as we update or steer parameters (controls - $\omega(t)$) affecting the latent state of the system, we can enforce constraints on the observable $d(t)$ by defining appropriate CBF $\mathcal{S}(d(t))$. We will adopt this framework to NN learning in the following sections.

## 3 PROPOSED METHODOLOGY

We aim to develop a learning approach for constraining the parameters using KGs with the same salient features mentioned in Section 2.2. This helps facilitate adherence to domain-grounding when NN parameters are initialized appropriately while offering an optimization framework to attain a stable solution. To formalize this idea, we treat the NN parameters as the control variable (analogous to $\omega$), domain model KGs define the constraints, the NN layer parameters form the latent states, and the observables are graph structured abstractions learned from the NN parameters. These are further elaborated in the following.

**KGs for Specifying Domain Models to Constrain NN Learning** To guide NN learning, we implement model constraints using knowledge graphs. Figure 2 illustrates how such a model for the drone example is defined using a KG. A concise explanation of the KG's representation is provided in the figure caption due to space concerns. These domain models form the constraints for the NN observables - the graph structured abstractions derived from NN parameters.

**Deriving Graph Structured Abstractions from NN Parameters** Given the excessive number of parameters in NNs, we leverage graphs derived from the latent parameters of the NN, similar to how the observed position of a drone is derived from its latent states. Using the ground-truth KGs, constraints can be defined to minimize the distance between the derived graph structure and a ground truth domain KG. Note that in this paper, we work with graphs where all edge relationships are of the same type (e.g., the precedes graph in Figure 2 that encodes preconditions for *causality* - which events precede one another). In the Section 5, we discuss ideas for extensions to multi-relational graphs and leave experimentation on this for future work. Our method works by emulating the learning process of a Graph Neural Network (GNN). A GNN learning procedure yields node representations characterized by two main features: (1) *Position Invariant Node Representations* - Node representations for the graph vertices do not model any notion of "node position" in the graph, i.e., they are position invariant, and (2) *Representations that Encode the Transitive Structure of the Graph* - in GNN methods, node representations aggregate neighborhood node information over multiple rounds resulting in encoding information from all nodes reachable through a transitive closure operation (in the limit). Now, you might wonder, *Why not directly utilize a GNN and instead emulate the GNN learning method?* Our proposition involves devising a solution that learns representations with the characteristics (1) and (2) within the forward pass of the overall NN learning process to address the downstream task at hand. We aim to integrate GNN learning seamlessly into the forward

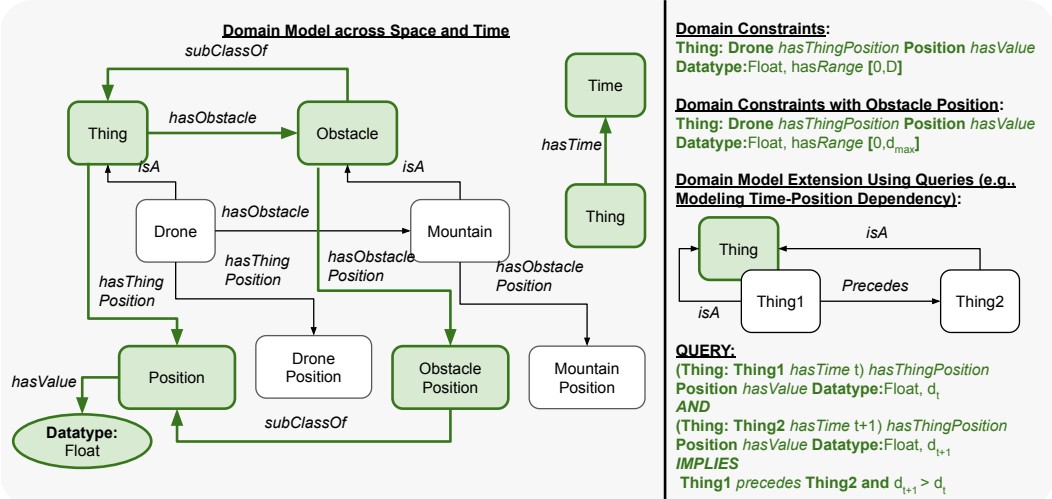

Figure 2: **Defining Domain Model for Drone Navigation Using a KG**. We employ a KG to specify our domain model and constraints in our Drone Navigation example context (see Figure 1). Within this framework, the drone is represented as a "Thing", and it's important to note that Things can encounter various "Obstacles". In our scenario, the primary obstacle is the mountain. To accurately depict the state of the drone, we incorporate two crucial aspects: "Time" and "Position". The drone exists at a specific point in "Time" and occupies a particular "Position" in space. Additionally, we can establish domain-specific constraints, such as defining the maximum allowable value for a thing's position. Moreover, we have the flexibility to expand our domain model by leveraging the KG querying capabilities. For instance, we can execute a SPARQL query on the OWL representation of the KG to introduce new elements and relationships. In the accompanying figure, we illustrate a query designed to introduce the concept of "precedence." This concept proves invaluable for modeling physical effects, including causality, where the direction of events in time is significant. Integrating such queries into our KG enhances our ability to represent and manage complex interactions within the domain. KGs are a natural choice for describing domain models and constraints in various real-world environments characterized by spatiotemporal dynamics. One prominent example is the representation of traffic scenes, where KGs capture intricate relationships and dependencies.

pass of our suggested training method, thus avoiding the need for an additional set of training loops to further train a conventional GNN within each epoch. Our experiments show that this modification improves overall accuracy (on the downstream task, e.g., next token prediction in Figure 3). This also boosts domain-grounding scores (compared against using GNNs as an inner loop, see Section 5). Figure 3 illustrates the graph node representation learning process that emulates the GNN learning process to achieve *position invariant node representations*, and *node representations that encode the transitive structure of the graph*. A detailed explanation of the process of deriving the proposed graph structured abstraction is given in the caption of the Figure 3.

**Devising a Solution Strategy for the Constrained Optimization Problem** Consider a generic optimization problem with the objective $\min_x f(x)$ such that $g(x) < 0$ holds. In the NN learning problem considered in this paper, the task-specific loss function could be specified using $f(x)$, and the domain constraints can be specified using $g(x)$. The decision variable $x$ is used to denote the NN parameters. This constrained optimization problem can be formulated as a soft-constrained optimization using a Lagrange multiplier $u$ so as to get $\min_x f(x) + u \cdot g(x)$, where $u$ can be interpreted as a penalty that is applied for violating the constraint $g(x)$. If the desired behavior is that the constraint is never violated, we can set $u = \infty$ if $g(x) > 0$ and $u = 0$ if $g(x) \leq 0$. The same thing can also be described as $\min_x f(x) + \max_u u \cdot g(x) = \min_x \max_u f(x) + u \cdot g(x)$. We now replace the $\max$ operation with the GumbelMax operator to obtain $\min_x \text{GumbelMax}_{u \in U} \{ f(x) + u \cdot g(x) \}$, where $U$ is some large and finite set of numbers (initialized randomly). This is a fully differentiable objective, i.e., end-to-end optimizable using gradient descent (GD) or stochastic GD (SGD). Thus, if $g(x)$ is a CBF, guaranteed to constrain $x$ to a forward invariant set during iterative optimization, then the solution converges to a stable solution because of the proven convergence guarantees of $SGD$ for NN learning for "well-behaved" objective functions (e.g., the cross-entropy

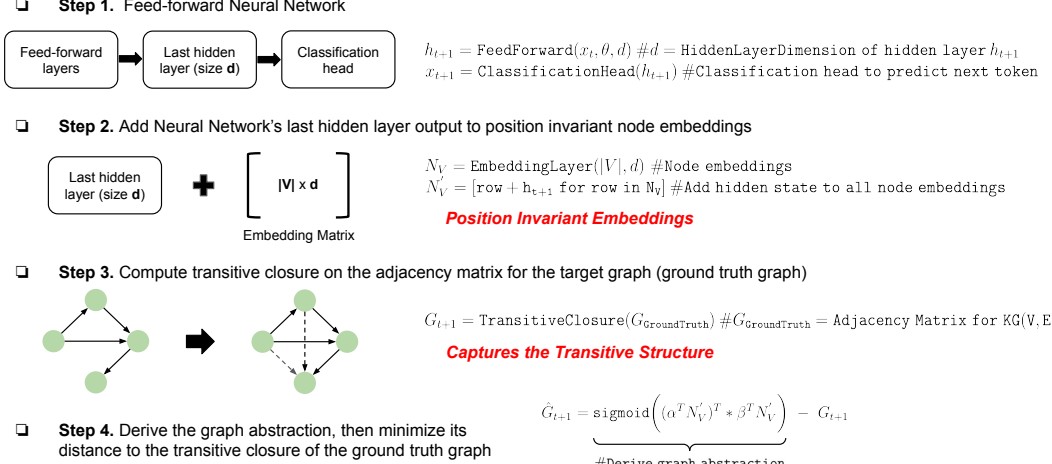

Figure 3: **Step 1.** Begin by constructing a feedforward neural network (NN) with parameters $\theta$ and input $x$. It's important to note that this structure has a flexible design. For instance, the feed-forward layers can be substituted with transformer-style encoders while the rest of the process remains consistent. **Step 2.** Create an embedding matrix with an embedding size denoted by $d$ for each of the $|V|$ nodes in the target graph. This ensures one embedding for each node. To simplify, ensure that the hidden layer size of the neural network matches the node embedding dimensions, both set to $d$. If there's a mismatch, an up-projection or down-projection layer is necessary. **Step 3.** Compute the transitive closure of the target graph, connecting nodes with a directed path between them on the graph. **Step 4.** Generate the graph abstraction using an "asymmetric inner product". This inner product involves $\alpha$ and $\beta$ to produce directed edges, differentiating it from a symmetric graph matrix. Optimize the learning of node representations by minimizing the distance between the derived graph abstraction and the ground truth graph, considering the transitive closure.

loss), while ensuring forward invariance (constraint is never violated if it is not violated during the start of the learning process).

To summarize, we train the NN models by defining the loss function $f(x)$ and the CBF - defined as the difference between the ground truth graph structure from domain-specific KG and the graph structures derived from NN parameters. To ensure that this constrained optimization problem is tractably solved, we reformulate this problem to get a soft-constrained optimization problem. Having covered the concepts related to the control of NNs with constraints, deriving graph structured abstractions from NN parameters, and the methodology for specifying spatiotemporal environment domain models and constraints through KGs, we illustrate how to integrate these features in Figure 4. This integration enables us to effectively achieve conformance to the constraints while maintaining forward invariance, obtaining a stable solution set, and ultimately facilitating domain-grounding. We design our experiments to rigorously evaluate this procedure on benchmark datasets using well-established metrics.

*Analysis:* We provide a theoretical analysis of the optimization problem (the last line of the pseudo-code in Figure 4). Specifically, we show convergence to the optimal parameter configurations using Stochastic Gradient Descent (SGD). *In summary, we provide proof that using our method approaches optimal downstream task performance while ensuring domain-grounding. The detailed proof is in the Appendix A. This is important because being equipped with the theory that the method works in the general case ensures that the reported results are not a dataset or task-specific artifact.*

## 4 DATASETS, TASKS, AND KGS

In this section, we refer to figures and their captions for elaboration due to space concerns.

### 4.1 DATASETS AND TASKS

Figure 5 provides a detailed illustration and explanation (in the caption) of the dataset and tasks we use for experimentation.

**Step 1. Transitive Structure + Forward Invariant CBF**

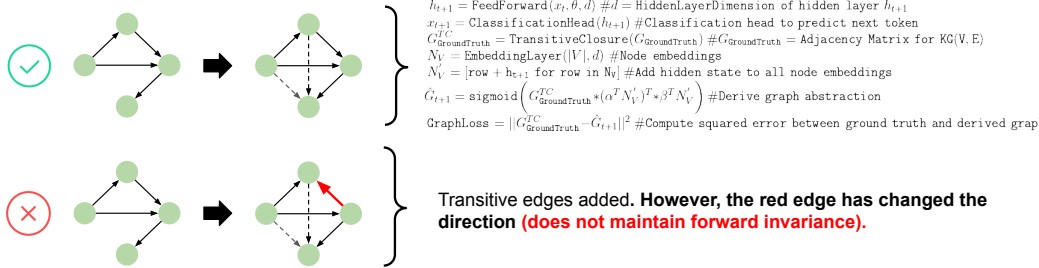

$$h_{t+1} = \text{FeedForward}(x_t, \theta, d) \text{ \#} d = \text{HiddenLayerDimension of hidden layer } h_{t+1}$$
$$x_{t+1} = \text{ClassificationHead}(h_{t+1}) \text{ \#Classification head to predict next token}$$
$$G^{TC}_{\text{GroundTruth}} = \text{TransitiveClosure}(G_{\text{GroundTruth}}) \text{ \#} G_{\text{GroundTruth}} = \text{Adjacency Matrix for KG}(V, E)$$
$$N_V = \text{EmbeddingLayer}(|V|, d) \text{ \#Node embeddings}$$
$$N_V' = [\text{row} + h_{t+1} \text{ for row in } N_V] \text{ \#Add hidden state to all node embeddings}$$
$$\hat{G}_{t+1} = \text{sigmoid}\left(G^{TC}_{\text{GroundTruth}} * (\alpha^T N_V')^T * \beta^T N_V'\right) \text{ \#Derive graph abstraction}$$
$$\text{GraphLoss} = ||G^{TC}_{\text{GroundTruth}} - \hat{G}_{t+1}||^2 \text{ \#Compute squared error between ground truth and derived graph}$$

Transitive edges added. **However, the red edge has changed the direction (does not maintain forward invariance).**

**Step 2. Stable Optimization Under Penalty Constraints**

$$\text{NextTokenLoss} = \text{CrossEntropy}(x_{t+1}, y_{t+1}), \text{ \#} y_{t+1} \text{ is the ground truth next token}$$
$$\text{GL}, \text{NTL} = \text{GraphLoss}, \text{NextTokenLoss } \text{\#Short hand notation}$$

$$\textbf{Minimize } \text{GumbelMax}([\text{NTL} + \lambda \cdot \text{GL} \;\forall\; \lambda]) \text{ \#differentiable penalty term optimization}$$

Figure 4: **Step 1.** We first derive the graph abstraction from the hidden layer output similar to Figure 3. However, we also multiply the graph abstraction with the transitive closure matrix. This acts as a mask that only allows learning of weights on edges that actually exist in the ground truth graph (thus maintaining forward invariance after each iteration, i.e., a domain-grounded solution to the parameters). **Step 2.** The GumbelMax reparameterization allows the optimization to discover how much penalty is incurred for violating the domain-grounding constraints during joint minimization of downstream task loss and constraint violation. This is important in order to arrive at a solution that boosts both downstream task performance and is domain-grounding scores (see evaluation section). Thus using **Steps 1.** and **2.**, we have proposed an end-to-end optimizable NN learning pipeline for domain-grounding with a KG.

- **CLEVRER** (20K videos, 300K questions)
  - 7 sec videos generated by physics engine for motion simulation
    - *Perfect* perceptual ground truth
    - 2 sec rendered, 2 sec held-out for prediction
  - Each question is paired with a functional program executable on video scenes

- **CLEVRER-HUMANS**
  - Events and causal relationships are annotated by humans with Mechanical Turk
  - Events are provided with textual descriptions; e.g. "The red cube hits the yellow ball"
  - 27 distinct event types (verbs in event text)
  - Causal effect estimation scores from 0 to 5

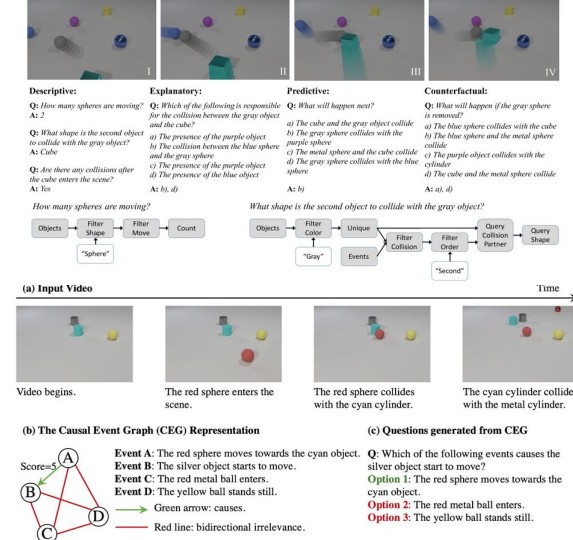

Figure 5: **CLEVRER dataset.** This dataset can be used for training a model to solve a question-answering task regarding spatiotemporal events between objects of different shapes, materials, and colors moving on a tabletop. The question categories are descriptive, explanatory, predictive, and counterfactual. The figure shows examples of the different categories. **CLEVRER-Humans**, an extension of **CLEVRER**, is also annotated with *Causal Event Graphs* (CEGs) that assign a score (0-5) on causes of events described using natural language. A crucial difference between the two, besides the CEGs, is that the answers in CLEVRER are functional program tokens (next token prediction = next program token prediction), and the answers in CLEVRER-Humans are natural language. In both domain-grounding of event sequence is necessary to answer the questions, thus making these datasets suitable for testing our method.

## 4.2 KGs

Similar to how the "precedes" graph is constructed in Figure 2, we construct a KG for the CLEVRER domain (with the schema elements objects, frames, videos, and object properties) and execute

queries on this KG to obtain a "precedes" graph that we call an *Event Ordering Graph* (EOG), that denotes which event happens before which other event in time. Figure 6 depicts and details this process.

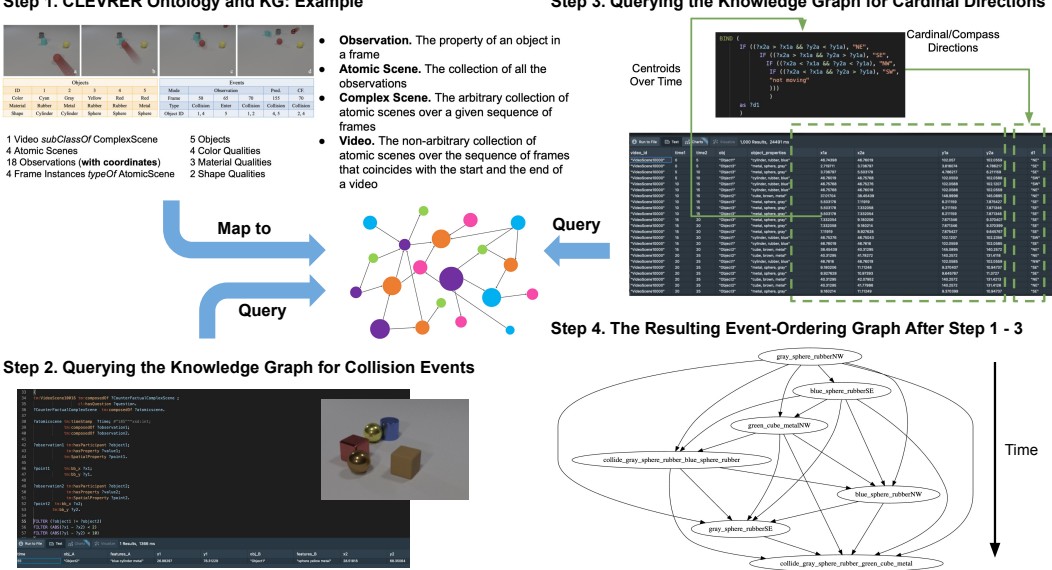

Figure 6: **Step 1.** Shows an example of how the frames in a video are mapped to the schema elements - Objects, colors, materials, frames, scenes, and shapes in a KG. **Step 2.** shows how the KG is queried to collision events. **Step 3.** shows how the KG is queried to get cardinal directions (north, east, west, south, and their combinations) of object trajectories across frames. Lastly, **Step 4.** shows the resulting "precedes" or *Event Ordering Graph*, which depicts which events happen before which other events in time.

## 5 EXPERIMENTS, RESULTS AND DISCUSSION

**Our method: NN + Optimization objective details:** We use a feed-forward network with two hidden layers of size 1000 for the NN that predicts the next token in the sequence. We use two embedding matrices before the feed-forward layers, one for embedding the tokens and a second for embedding the positions of the tokens in the sequence. We use an embedding size of 96. We experiment with the SGD optimizer with a learning rate of 0.01 and the AdamW optimizer. We randomly initialize the list of $\lambda$ (the set $U$ in Section 3) and set the list size to 100 [1].

**Graph Node Representation Learning Experiments:** We experiment with state-of-the-art (SOTA) graph node representation learning (GNRL) techniques - TransE, DistMult, CompIEx, HoIE (ampligraph implementation) trained on the EOGs, and report the performance using standard link prediction metrics (link exists if the cosine similarity between node representation vectors is greater than 0.5).

**Baseline Method** For a baseline implementation, we use the code almost exactly based on the pseudocode in Figure 4 with the additional step of adding the average node embeddings from the SOTA GNRL techniques to the rows of $\mathbb{N}_V$ when obtaining $\mathbb{N}'_V$. We chose this as our baseline as the traditional method of incorporating graph information in NN learning pipelines is through the use of GNRL embeddings integrated with the NN's internal representations. However, we also provide the leaderboard results in Section B.1 for comparison against SOTA methods on the CLEVRER benchmark. Figure 7 details the results. We ask the reader to refer to the figure caption for a detailed explanation of the results and the discussion due to space concerns.

---

[1]Limited Code and KG provided as a zip file, full code will be made available upon acceptance

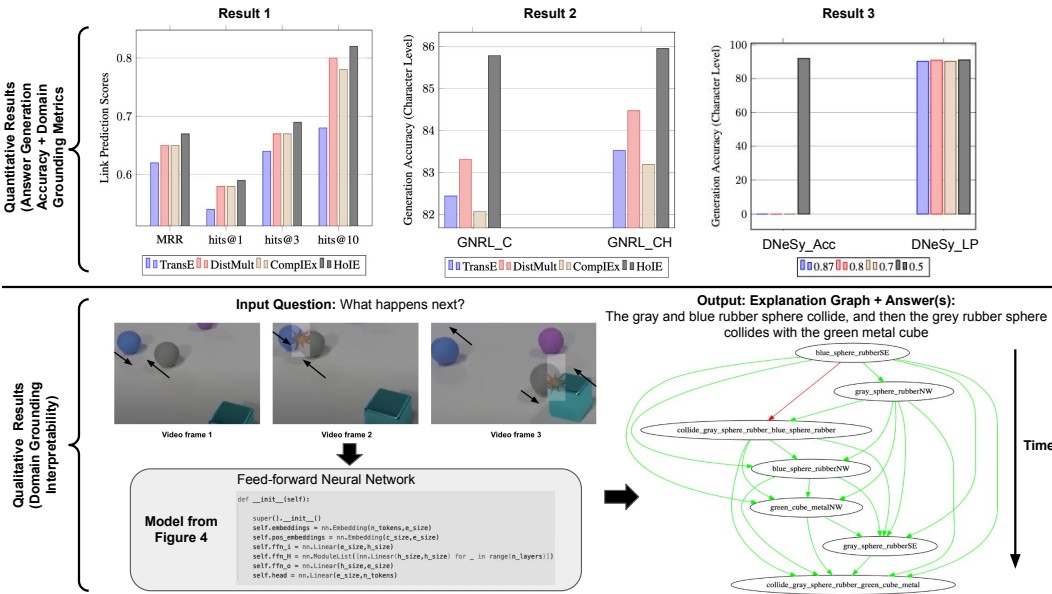

Figure 7: *Quantiative Results:* All the results are averaged across all the videos and question categories in the test set. **Result 1.** shows that the SOTA techniques perform subpar across traditional link prediction metrics except hits@10 after training on the training EOGs. **Result 2.** shows the results of the baseline implementation - GNRL_C refers to results on the CLEVRER dataset, and GNRL_CH refers to results on the CLEVRER-Humans dataset. The result shows the accuracy of answer token generation compared to the ground truth. **Result 3.** (left) shows the answer generation accuracy of our method denoted by DNesy (stands for domain-grounded neural system), and the right shows link prediction results for hits@1. We see that the accuracy of text generation is better using our method vs. the baseline for both datasets (see Section B.1 for additional context on the numbers). The numbers in the legend refer to the different thresholds used for link prediction using our method. We check if the corresponding entry in $\texttt{sigmoid}(\alpha^\mathrm{T}\mathrm{N}_\mathrm{V}')^\mathrm{T} * \beta^\mathrm{T}\mathrm{N}_\mathrm{V}'$ (see Figure 4) is greater than the threshold. Our method's link prediction results are significantly better than the SOTA methods across all thresholds, thus providing compelling evidence of domain-grounding using our method. *Qualitative Results:* The figure shows a complete execution snapshot - the input to the model is the question and the EOG corresponding to the video. The output is the generated answer and an explanation graph, where edges in the input EOG are colored green if the corresponding entry in $\texttt{sigmoid}(\alpha^\mathrm{T}\mathrm{N}_\mathrm{V}')^\mathrm{T} * \beta^\mathrm{T}\mathrm{N}_\mathrm{V}'$ is greater than the threshold, else red. Ideally, all edges should be green for perfect domain-grounding as all those edges are true precede relations in the actual video. Please refer to Section B.2 for more output examples.

## 6 CONCLUDING REMARKS

**Conclusion** We proposed a method for domain-grounding in NN learning using a KG to specify the domain model and constraints. Furthermore, we provide a theoretical analysis of the solution to our proposed objective. Our results demonstrated the efficacy of the proposed approach on established benchmark datasets using various quantitative and qualitative analysis metrics.

**Future Work** We plan to implement multi-relational KGs by deriving graph abstractions as three-dimensional tensors (instead of two-dimensional matrices). We also plan to explore learning systems with interconnected subsystems, each with its own constraint specification. Finally, we plan to perform more comprehensive experiments on real-world datasets (e.g.,(Yao et al., 2020)).

**Broader Impact** In real-world interconnected systems, the individual subsystem constraints may correspond to different social aspects, e.g., safety, ethics, social constructs, legal rules, etc. Large NN-based systems that model various social aspects are slowly being adopted across safety-critical industries with humans in the loop. Our work provides concrete steps towards a principled method to provide adherence guarantees and enhanced explainability of outcomes.

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

# A    APPENDIX

**Analysis of the learning rule and the impact of the hyper-parameters on constraints and convergence**    In the following, we show how the standard SGD can be used to optimize our proposed objective. We include this for completeness.

[2] Consider the optimization problem of finding the optimal NN parameters

$$\theta^* = \arg\min_\theta f(\theta) + \lambda g(\theta). \tag{4}$$

We use the SGD update rule given by

$$\theta_{t+1} = \theta_t - \delta_t \cdot \nabla f(\theta_t) + \lambda \nabla g(\theta_t),$$

where $\delta_t$ is the learning rate, and $\nabla$ denotes the gradient of the function with respect to its parameters. First, we write the SGD formula as follows:

$$\frac{\theta_{t+1} - \theta_t}{\delta_t} = -\nabla(f(\theta) + \lambda g(\theta))$$

Here, $g$ is the squared distance between the graph abstraction (a matrix) and the transitive closure applied on the ground truth graph (an adjacency matrix), and $\lambda$ is a penalty that is proportional to this distance. This can be seen as a finite difference approximation of the derivative of the continuous function $f(\theta) + \lambda g(\theta)$, i.e., a discretization of the ordinary differential equation, for sufficiently small $\delta_t$, given by

$$\dot\theta_t = -\nabla(f(\theta_t) + \lambda g(\theta_t))$$

We now define an energy function and show that this energy is a control Lyapunov function. Finally, we bound the energy and show convergence.

We define the candidate Lyapunov function $E(\theta_t, \theta^*)$ as $f(\theta_t) + \lambda g(\theta_t)$, where $f$ is the task-specific loss, e.g., the cross-entropy loss or squared-error loss. Three out of four properties of a Lyapunov, i.e., (1) $E$ is continuous, (2) $E(\theta_t) = 0$ if and only if $\theta_t = \theta^*$, and (3) $E(\theta_t) > 0$ if and only if $\theta_t \neq \theta^*$ trivially hold. (1) Because $E$ is a sum of continuous functions, (2) and (3) because of the definition of the functions $f$ and $g$. Now we need to prove the fourth property, which says that $E(\theta_{t+1}) \leq E(\theta_t)$, $\forall t$.

We assume that for some constants $M > 0$ and $L > 0$, $||\nabla(f(\theta_t) + \lambda g(\theta_t))|| \leq M$ and $|u^T \nabla^2(f(\theta_t) + \lambda g(\theta_t))u| \leq L||u||^2$. This is a natural assumption for a discrete algorithm implemented using a finite dataset. From Taylor's theorem, there exists a $\eta_t$ such that

$$
\begin{aligned}
f(\theta_{t+1}) + \lambda g(\theta_{t+1}) &= f(\theta_t - \delta_t \nabla f(\theta_t)) + \lambda(g(\theta_t - \delta_t \nabla g(\theta_t)) \\
&= \Big( f(\theta_t) + \lambda g(\theta_t) - (\delta_t \nabla(f(\theta_t) + \lambda g(\theta_t)))^T \nabla(f(\theta_t) + \lambda g(\theta_t)) \\
&\quad + \frac{1}{2}(\delta_t \nabla(f(\theta_t) + \lambda g(\theta_t)))^T \nabla^2(f(\eta_t) + \lambda g(\eta_t))(\delta_t \nabla(f(\theta_t) + \lambda g(\theta_t))) \Big) \\
&\leq \Big( f(\theta_t) + \lambda g(\theta_t) - (\delta_t \nabla(f(\theta_t) + \lambda g(\theta_t)))^T \nabla(f(\theta_t) + \lambda g(\theta_t)) \\
&\quad + \frac{\delta_t^2 L}{2} ||\nabla(f(\theta_t) + \lambda g(\theta_t))||^2 \Big) \\
&\leq \Big( f(\theta_t) + \lambda g(\theta_t) - (\delta_t \nabla(f(\theta_t) + \lambda g(\theta_t)))^T \nabla(f(\theta_t) + \lambda g(\theta_t)) \\
&\quad + \frac{\delta_t^2 L M^2}{2} \Big)
\end{aligned}
$$

---

[2]The analysis presented here is based on `https://www.cs.cornell.edu/courses/cs4787/2019sp/notes/lecture5.pdf`

As mentioned earlier, SGD is implemented with a finite dataset. Therefore, we can take the expectation (expectation with respect to the randomness in the dataset) on both sides to get:

$$\mathbb{E}[f(\theta_{t+1}) + \lambda g(\theta_{t+1})] \leq \mathbb{E}[f(\theta_t) + \lambda g(\theta_t)] - \delta_t \mathbb{E}[||\nabla(f(\theta_t) + \lambda g(\theta_t))||^2] + \frac{\delta_t^2 L M^2}{2}$$

Rearranging and summing up over $T$ iterations, we get:

$$\sum_{t=0}^{T-1} \delta_t \mathbb{E}[||\nabla(f(\theta_t) + \lambda g(\theta_t))||^2] \leq \left( \sum_{t=0}^{T-1} (\mathbb{E}[f(\theta_t) + \lambda g(\theta_t)] - \mathbb{E}[f(\theta_{t+1}) + \lambda g(\theta_{t+1})]) \right.$$
$$\left. + \sum_{t=0}^{T-1} \frac{\delta_t^2 L M^2}{2} \right)$$
$$\leq (f(\theta_0) + \lambda g(\theta_0)) - (f(\theta^*) + \lambda g(\theta^*)) + \frac{L M^2}{2} \sum_{t=0}^{T-1} \delta_t^2$$

$f(\theta^*) + \lambda g(\theta^*)$ is an optimal solution. Note that all the analysis until now still holds if $\lambda$ is wrapped in a differentiable objective (e.g., `GumbelMax` over a finite set $U$), where the function $f(\theta) + \lambda g(\theta)$ can be replaced with a new function $F(\Theta)$, where $\Theta = [\theta, U]$. Thus, we see that in expectation over $T$ iterations, $f(\theta) + \lambda g(\theta)$ approaches $f(\theta^*) + \lambda g(\theta^*)$, i.e., the fourth property of the Lyapunov $E$ holds. This means using our proposed method, an optimal $\theta$ and $\lambda$ configuration that ensures optimal domain-grounding is achieved using SGD. This configuration is achieved for any given NN loss function $f$ and a constraint term $g$ representing the proximity to a KG, where the proximity is influenced by the parameters $\lambda$ and $\theta$.

# B APPENDIX - ADDITIONAL QUALITATIVE RESULTS AND LEADERBOARD RESUTLS

## B.1 LEADERBOARD RESULTS

| Model | Average-↑ per ques. | Descriptive | Explanatory- per opt. | Explanatory- per ques. | Predictive- per opt. | Predictive- per ques. | Counterfactual- per opt. | Counterfactual- per ques. |
|-------|---------|-------------|----------|----------|----------|----------|----------|----------|
| AI Core | 95.24 **91.85** | 96.46 | 99.94 | 99.81 | 93.96 | 93.96 | 96.61 | 90.72 |
| redherring | 91.14 | 95.76 | 98.88 | 96.98 | 95.69 | 91.75 | 92.97 | 80.05 |
| VRDP | 90.24 | 93.4 | 96.3 | 91.94 | 95.68 | 91.35 | 94.83 | 84.29 |
| Fighttttt | 88.71 | 94.77 | 98.25 | 95.46 | 94.16 | 89.25 | 91.25 | 75.35 |
| neural | 88.27 | 94.01 | 98.47 | 95.99 | 93.49 | 87.48 | 91.42 | 75.61 |

Figure 8: Leaderboard results from the CLEVRER dataset for context when viewing results reported in this paper, across all question categories (see Section 5). Our method's score of 91.85 is highlighted in yellow.

## B.2 APPENDIX - ADDITIONAL QUALITATIVE RESULTS WITH EXAMPLES FROM CLEVRER AND CLEVRER-HUMANS

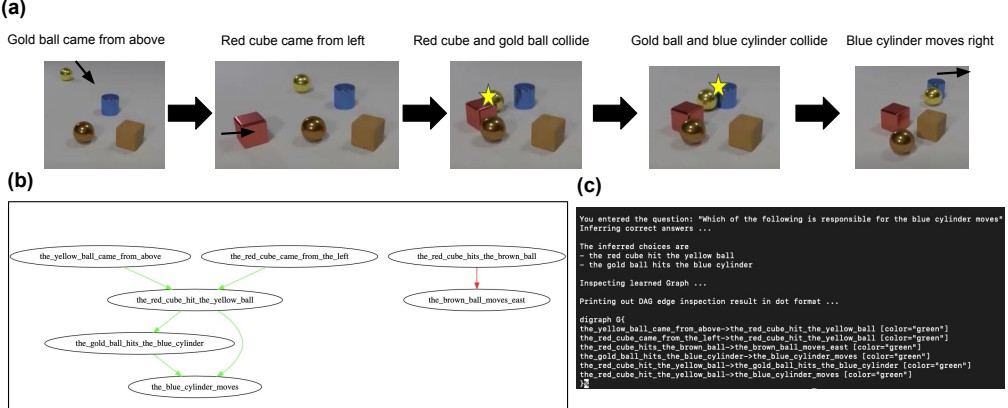

Figure 9: An example output from the CLEVRER-Humans dataset. **(a)** We see that first, the golden ball enters from the top, the red ball then moves in from the left, and the two collide. This, in turn, causes the golden ball to hit the blue cylinder, moving right. **(b)** Shows the output explanation graph depicting which events happened in what ordering in time, where the green edges correspond to the quantity $\texttt{sigmoid}(\alpha^{\mathtt{T}}\mathtt{N}_{\mathtt{V}}')^{\mathtt{T}} * \beta^{\mathtt{T}}\mathtt{N}_{\mathtt{V}}'$ crossing a threshold. **(c)** Shows the output of the NN, which generates natural language text corresponding to the question - "Which of the following is responsible for the blue cylinder moves". The NN answers correctly as *the red cube hit the yellow ball*, and *the gold ball hit the blue cylinder*. Note that the NN interchanges yellow and gold, presumably due to the dataset reflecting either token as having similar statistical associations to other tokens.

