# OpenReview forum: "DOMAIN-GROUNDING OF NEURAL NETWORKS FOR SPATIOTEMPORAL REASONING"
_ICLR.cc/2024/Conference — ICLR 2024 Conference Withdrawn Submission_

### Official Review · Reviewer_g76t · 2023-10-26

**Soundness:** 2 fair
**Presentation:** 1 poor
**Contribution:** 2 fair
**Rating:** 3
**Confidence:** 2

**Summary:**

The authors propose a control theoretic perspective for learning with knowledge graphs that represents domain models and constraints. Their proposed deep learning approach constrains parameters with these knowledge graphs. The paper includes theoretical analysis and experiments on the CLEVRER and CLEVERER-Humans datasets, which focuses on question answering in videos.

**Strengths:**

The broad idea of training networks whose parameters abide by ground truth graph structures from domain-specific knowledge graphs is clever. I appreciate the author’s analysis on how the stated constrained optimization problem can be reformulated to be a soft-constrained problem, though I cannot provide feedback on the theory parts of the paper.

**Weaknesses:**

W1. I am quite confused at the paper formulation and key points. I’m not quite sure how this very in-depth drone example is relevant to constraining NNs where constraints are specified as knowledge graphs. I also don’t understand what part of this is “domain-grounding”. What is being grounded to the domain?

W2. It seems like deriving graph structured abstractions from NN parameters is a core part of the paper, but I don’t understand the authors’ motivation for not using GNNs, and instead proposing the complex method in Figure 3. Are there ablations to show that this is important?

W3. Is the knowledge graph given in train and in test? The training and test paradigm is very unclear. What information is given?

W4. This also seems constrained to knowledge graphs where all edge relationships are the same; usually knowledge graphs depict different relations in the edges. In the drone example given this seems to be the case.

W5. What are the ground truth graphs for CLEVRER and CLEVRER-Humans? How is the knowledge graph queried to collision events? Can you give more examples of “domain models”?

Nit: The figures are also quite confusing.

**Questions:**

See above.

---

### Official Review · Reviewer_f8Mp · 2023-10-31

**Soundness:** 1 poor
**Presentation:** 1 poor
**Contribution:** 1 poor
**Rating:** 1
**Confidence:** 3

**Summary:**

The authors propose a solution that incorporates symbolic representation (knowledge graphs) for domain understanding into the architecture of neural networks. The approach is inspired by control barrier function from control theory and aims to impose domain knowledge-graph constraints during training. The effectiveness of the approach is demonstrated using two benchmark datasets: CLEVRER and CLEVRER-Humans, which focus on spatiotemporal reasoning and question answering.

**Strengths:**

The paper presents an interesting idea of enforcing knowledge-graph-derived constraints in neural network training. It helps better recover the graphical structures underlying physical trajectories (e.g., collision event graphs).

**Weaknesses:**

The paper presentation is much below the quality requirment of the conference. For example, the paper spent a lot of effort describing drone dynamics and knowledge graphs for drone navigation, which is completely irrelevant to the experiments. Furthermore, there is no clear description of what's exactly the form of the KGs used in the experiments, what's the prediction task of the KG; etc.

All the experimental results are compared with transductive KG methods, which I don't understand how they are related to the CLEVRER QA tasks...

**Questions:**

What are those pictures in Figure 6 (KGs, KG querying software)? I can't find the relevant code in the supplementary zip.

What's the difference between the CBF-inspired formulation and just auxiliary training losses? For example, Embedding Symbolic Knowledge into Deep Networks https://proceedings.neurips.cc/paper/2019/hash/7b66b4fd401a271a1c7224027ce111bc-Abstract.html

---

### Official Review · Reviewer_vq2f · 2023-11-01

**Soundness:** 3 good
**Presentation:** 2 fair
**Contribution:** 3 good
**Rating:** 3
**Confidence:** 3

**Summary:**

In this paper, the authors propose a framework for integrating domain knowledge (expressed with knowledge graphs) into neural networks. The main idea is to benefit from concepts in control theory and design a differentiable objective that constrains model predictions within the domain of allowed parameters. In doing this, the authors introduce the notion of control barrier function (CBF) and extend it to neural models where the graph resulting from its representation is compared to the ground-truth knowledge graph. The experiments are conducted over two spatiotemporal reasoning benchmarks: CLEVRER and CLEVRER-Humans, where the model attains better performances w.r.t. standard GNNs competitors and it is highly performant within the leader board of CLEVRER.

**Strengths:**

Merging neural processing with symbolic information is a hot topic within Neuro-Symbolic AI that has potential benefits for improving standard neural computing. Knowledge graphs are well known in the community, as well as tools for handling them, and merging them within neural networks is a promising research area.

The authors' idea to make use of control theory is new and deserves attention as a principled way to impose constraints in the model. One key property is the forward invariance that guarantees that once constraints are satisfied, there is forward compliance with them. This is interesting and the analysis of the optimization is also included. Furthermore, the injection of the knowledge graph evidences improvements to neural competitors.

**Weaknesses:**

The presented material is of high quality but misses citing relevant works in the field that are more aligned to the applications of knowledge in learning. In particular, the study in knowledge grounding is quite (seen as NeSy AI) is quite progressed, see [1], as well as works integrating GNNs in vision, see [2]. It should be clear why the baselines chosen represent the SotA and it should be mentioned clearly what are key differences of the proposed method. In particular, semantic regularization losses were proposed both in fuzzy and probabilistic logic, see [3,4].

The paper presents the intuition on control theory in an intuitive manner but lacks further details that could be useful for a more thorough comprehension of the contribution. In particular, devoting all details of the model in Figure 3 and other explanations in the figures renders the presentation quite heavy and some details are not entirely clear or not in evidence enough.

It is written that the nodes of the predicted graph must match the ground truth but it is not clear how it is done with the "asymmetric inner product". What are $\alpha$ and $\beta$? How are the nodes of the graph derived from the input $\mathbf x$ and from its last hidden layer? Is the implementation of the model sensible to the initial choice of the ground truth graph?

The authors introduce the fact that forward compliance with the constraints can be guaranteed upon finding initial states that satisfy the constraints. This is however hard to impose at the beginning of the learning and it leaves open how and whether the model would converge to be compliant. It is more realistic not to be entirely compliant, but it is not discussed what are the implications. How does the model behave in those scenarios?

[1] De Raedt, Luc, et al. "From statistical relational to neuro-symbolic artificial intelligence." arXiv (2020) \
[2] Senior, Henry, et al. "Graph Neural Networks in Vision-Language Image Understanding: A Survey." arXiv (2023) \
[3] Diligenti, Michelangelo, Marco Gori, and Claudio Sacca. "Semantic-based regularization for learning and inference." AI (2017) \
[4] Xu, Jingyi, et al. "A semantic loss function for deep learning with symbolic knowledge." ICML (2018) \

**Questions:**

I understood that the proposed method also applies to context beyond spatio-temporal reasoning. Is that the case? What would be the comparison with static image reasoning tasks like semantic image interpretation [5]?

Other questions are asked in the previous sections.

[5] Donadello, Ivan, Luciano Serafini, and Artur D'Avila Garcez. "Logic tensor networks for semantic image interpretation." arXiv (2017).

### Update on review

After seeing other reviewers' replies, I changed my vote to reject due to the quality of the presented material. I align with the claims that the paper could have been presented in a better way and the experimental description is somehow confusing.

---

### Official Review · Reviewer_1Yru · 2023-11-03

**Soundness:** 3 good
**Presentation:** 2 fair
**Contribution:** 2 fair
**Rating:** 3
**Confidence:** 4

**Summary:**

Inspired by dynamical systems, the authors propose a constraint on the internal features of a neural network. In particular, they design a knowledge graph with constraints focused towards spatiotemporal reasoning. They then show that thus constraining the learning process leads to better domain grounding (i.e. stronger reliance on the underlying data generative process). The authors also propose an interpretability score to quantify how strongly the learned constraints are followed for a particular reasoning instance.

**Strengths:**

The analogy drawn to dynamical system constraints and neural network inductive biases, and its use define a novel KG driven learning constraint is a clever modelling choice for spatiotemoral reasoning.

**Weaknesses:**

* The authors define domain grounding as "grounding in reality or truth". Then they mention in "Related Work" that domain grounding in NNs has been attempted via in-context learning. This is a very limited view of prior work on grounding in neural networks. I think the authors need to include more references/discussion on prior work on grounding them on external structured/unstructured data: visual scenes [1], external knowledge [2], information retrieval [3] etc.

* "These constraints typically do not form an intrinsic component of the...model; instead, they arise from environmental or domain-specific information." The authors work links to the work huge body of work in neural network learning and reasoning involving inductive biases that do exactly what the authors seek to do - invoke constraints on the model, the data, or the learning process that reflect domain-specific information [4]. In particular, introducing problem based constraints on the hidden representation. Beyond the dynamical systems analogy, how is this sort of constraining on the hidden representation by introducing a constrained loss function component anything novel?

* I fail to see how such a method translates/scales to a different domain. Defining the domain specific KG constraint model would involve strong human priors informed by the domain knowledge. Any solution is by definition rendered to be highly domain specific and limited.

**References**

1. Wang, P., Wu, Q., Shen, C., Dick, A. and Van Den Hengel, A., 2017. Fvqa: Fact-based visual question answering. IEEE transactions on pattern analysis and machine intelligence, 40(10), pp.2413-2427.
2. Ji, Z., Liu, Z., Lee, N., Yu, T., Wilie, B., Zeng, M. and Fung, P., 2023, July. Rho: Reducing hallucination in open-domain dialogues with knowledge grounding. In Findings of the Association for Computational Linguistics: ACL 2023 (pp. 4504-4522).
3. Lewis, P., Perez, E., Piktus, A., Petroni, F., Karpukhin, V., Goyal, N., Küttler, H., Lewis, M., Yih, W.T., Rocktäschel, T. and Riedel, S., 2020. Retrieval-augmented generation for knowledge-intensive nlp tasks. Advances in Neural Information Processing Systems, 33, pp.9459-9474.
4. Goyal, A. and Bengio, Y., 2022. Inductive biases for deep learning of higher-level cognition. Proceedings of the Royal Society A, 478(2266), p.20210068.

**Questions:**

Since a lot of the important modelling details and derivations are currently in figure captions, would it be possible for the authors to reorganize their paper with a supplementary section to improve readability?

---

### Meta-Review · Area_Chair_Kfd4 · 2023-12-04

**Metareview:**

Reviewers were unanimous in their ratings and reviews that this paper was significantly below acceptable quality, and the authors did not post any responses at all to any of the reviews.

Overall, reviewers agreed that the paper was poorly written and very confusing, with the example of drones being cited by multiple reviewers as unclear how it relates to "domain grounding".

More specifically, multiple reviewers pointed out that the paper's use of "grounding" was very confusing and seemed to be different from the standard notion of grounding in computer vision. Other major issues include those related to evaluation, such as unclear training and test paradigms.

In summary, this paper was found to be simply hard to understand, poorly communicated in multiple ways, and clearly not suitable for ICLR acceptance.

**Justification For Why Not Higher Score:**

Simply a low-quality paper that is clearly not suitable for ICLR, as justified in the meta-review.

**Justification For Why Not Lower Score:**

N/A

---

### Decision · Program_Chairs · 2024-01-16

Reject